# Enhancing Event Camera Data Pretraining via Prompt-Tuning with Visual Models

## Abstract

The pretraining-finetuning paradigm has achieved remarkable success in natural language processing and computer vision, becoming the dominant approach in many downstream tasks. However, its application in the event camera domain has encountered significant challenges. First, the scarcity and sparsity of large-scale event datasets lead to issues like overfitting during extensive pretraining. Second, event data inherently contains both temporal and spatial information, making it difficult to directly transfer knowledge from image-based pretraining to event camera tasks. In this paper, we propose a low-parameter-cost SpatioTemporal Information Fusion Prompting (STP) method to address these challenges. This method enables bidirectional fusion of event and image data while mitigating the risk of overfitting. Specifically, the key innovation lies in effectively integrating the spatio-temporal information of event data to align with pre-trained image models and reduce the impact of data sparsity. To achieve this, we designed an Overlap Patch Embedding module within the STP, which employs wide receptive field to capture more local information and reduce the influence of sparse regions. Additionally, we introduce a Temporal Transformer that integrates both global and local information, facilitating the fusion of temporal and spatial data. Our approach significantly outperforms previous state-of-the-art methods across multiple downstream tasks, including classification, semantic segmentation, and optical flow estimation. For instance, it achieves a top-1 accuracy of **68.83%** on N-ImageNet with fewer trainable parameters. Our code is available in the **Supplement**.

## 1 Introduction

Event cameras are dynamic vision sensors inspired by the perceptual mechanism of the human retinas Lichtensteiner et al. (2008); Taverni et al. (2018); Schuman et al. (2022). They asynchronously capture the event stream by comparing the intensity changes of each pixel Taverni et al. (2018); Gallego et al. (2020); Brandli et al. (2014). These events are sorted as positive or negative depending on whether the light intensity increases or decreases Lichtensteiner et al. (2008); Gallego et al. (2020); Chen & Guo (2019). This triggering mechanism enables event cameras to efficiently record information in high-dynamic-range (HDR, 120dB) or high-speed motion scenes, while offering advantages such as low power consumption and low redundancy Lichtensteiner et al. (2008). Currently, event cameras are widely used in novel computer vision and robotics tasks, including video interpolation Tulyakov et al. (2022); Yu et al. (2021); Gao et al. (2022); Sun et al. (2023), image or video reconstruction Rebecq et al. (2019); Paredes-Vallés & De Croon (2021); Munda et al. (2018); Simon Chane et al. (2016), optical flow estimation Zhu & Yuan (2018); Lee et al. (2020), depth estimation Zhu et al. (2019); Gallego et al. (2018), detection Ramesh & Yang (2020); Ramesh et al. (2020), and SLAM Vidal et al. (2018); Mueggler et al. (2017); Jiao et al. (2021).

However, due to the high cost of acquiring event camera data and the difficulty of labeling, there is still a lack of pretrained models based on large-scale event camera datasets. This has hindered the adoption of the pretraining-finetuning paradigm for event-based vision tasks and limited the development of corresponding deep learning methods and models. Given the emergence of large-scale RGB image datasets (e.g., ImageNet-21k Deng et al. (2009), JFT-300M Sun et al. (2017)) and the development of pretrained models He et al. (2016); Dosovitskiy et al. (2020); Radford et al. (2021); He et al. (2022) based on these datasets, researchers have attempted to use transfer learning Hu et al. (2020); Sun et al. (2022); Messikommer et al. (2022) or knowledge distillation Wang et al.

Figure 1: **Comparison of different pretraining methods for event data.** Previous methods convert event streams into 2D representations, leading to the loss of temporal information. Our method preserves both spatial and temporal information by converting event streams into STECM and utilizing STP for bidirectional knowledge transfer between the event data and the image.

(2021a) to transfer the knowledge from RGB image-trained models to event-based downstream tasks. This approach has shown some success in smaller-scale tasks like semantic segmentation Sun et al. (2022), as event data and RGB images share similar object edge information. However, it struggles to generalize to other tasks, such as classification or optical flow estimation, because event data lacks color and texture information, while RGB images lack temporal information.

To address the lack of event-based pretrained models, some researchers have attempted pretraining on the N-ImageNet dataset Kim et al. (2021). They have demonstrated that the pretraining-finetuning paradigm is effective on event camera data as well Yang et al. (2023); Klenk et al. (2024). The biggest limitation of these methods is that they stack event streams onto a 2D plane to form event images (Figure 1), effectively ignoring the temporal information inherent in event streams. Additionally, the largest event camera dataset currently available is only obtained from ImageNet-1K Deng et al. (2009), which still presents a significant gap compared with large-scale image datasets like ImageNet-21k or JFT-300M. Coupled with the sparsity of event data, this imbalance between the extensive number of trainable parameters and the limited data can lead to overfitting in pretrained models.

In this paper, **we identify three key challenges for event camera data pretraining**: (i). The need to account for both temporal and spatial information in event data. (ii). The need to handle the sparsity and noise inherent in event data. (iii). The need for sufficient prior knowledge to mitigate the imbalance between the large number of trainable parameters and the limited available data.

To address these challenges, we propose a method that transfers knowledge from the image domain to the event data domain through appropriate prompting engineering. This approach not only aggregates the temporal and spatial information of event data and reduces the impact of sparsity, but also effectively transfers knowledge from the image domain to the event data domain, mitigating the issue of overfitting. Specifically, we design a SpatioTemporal Information Fusion Prompting (STP) method, which uses wide-receptive-field overlapping convolutions combined with a temporal transformer to gradually fuse the temporal and spatial information of event streams. This reduces the impact of data sparsity and produces a spatiotemporal representation of the event data. Then the representation is fed into a pretrained image model, where the model's weights are frozen to extract high-level features from the event data for classification. Through end-to-end optimization, the image-domain prior knowledge guides the training of the prompting module, enhancing spatiotemporal information fusion and facilitating knowledge transfer. Finally, STP can be combined with the pretrained image model, forms a pretraining model for event camera data, which is finetuned together on downstream tasks. This enables bidirectional knowledge flow between event data and image data and generalizes well to a variety of downstream tasks.

In summary, our work makes the following contributions:

- We propose a novel event camera pretraining method based on prompt-tuning, which facilitates bidirectional knowledge fusion between event data and image data through spatiotemporal feature fusion prompting. This approach drives a new pre-training paradigm for event-based vision tasks.

- We propose a representation that preserves both the spatial and temporal information of event streams and introduce a Spatiotemporal Information Fusion Prompting method, specifically designed to gradually integrate the spatial and temporal features of event data, effectively addressing its unique characteristics.

- Our method achieves state-of-the-art (SOTA) performance across downstream tasks such as classification, semantic segmentation, and optical flow estimation. For instance, we achieve a top-1 accuracy of **68.83%** on the N-ImageNet dataset.

## 2 RELATED WORK

### 2.1 VISUAL PRE-TRAINING

With the rapid development of deep learning and computer vision, pretrained large models have become an important topic and research method Hendrycks et al. (2019); Raghu et al. (2021); He et al. (2019), driven by the continuous evolution of visual models. From the perspective of training, these methods mainly include supervised pre-training on large-scale datasets Carreira & Zisserman (2017); Dosovitskiy et al. (2020); Zhai et al. (2022); Dehghani et al. (2023), weakly supervised pre-training requiring less data Berthelot et al. (2019); Pham et al. (2021); Xie et al. (2020); Zheng et al. (2021); Ramanathan et al. (2021), and unsupervised pre-training that exploits intrinsic features of data for learning without relying on any labels Bao et al. (2021); Chen et al. (2020a;b); He et al. (2022); Grill et al. (2020). These models can efficiently transfer knowledge to downstream tasks through tuning.

In contrast to the rapid development of image-based pre-training, event-based pre-training is still in its early stages. There are two main challenges in pre-training with event camera data: first, the difficulty in acquiring event stream data and the lack of large-scale datasets; second, the sparsity of event stream data, which easily leads to overfitting or training collapse during large-scale training. Previous pre-training methods have been primarily relied on self-supervised learning. Yang et al. Yang et al. (2023), were the first to propose a contrastive learning-based method for large-scale event camera data pre-training. Klenk et al. Klenk et al. (2024), drew inspiration from VQVAE's discrete encoding Van Den Oord et al. (2017) and BERT's masked reconstruction Devlin et al. (2018) to propose Masked Event Modeling (MEM). Huang et al. Huang et al. (2024), introduced an efficient self-supervised learning method based on voxel-based data, enabling rapid convergence with a small amount of pre-training data. However, these methods are still limited by the lack of large-scale datasets and the sparsity of event data. Additionally, they do not fully consider the temporal characteristics of event data, which hinders the model's ability to fully leverage its learning and representation capabilities.

### 2.2 PROMPT TUNING

Prompt tuning is an important paradigm that leverages pretrained large models Lester et al. (2021); Liu et al. (2023). As a lightweight tuning method, its principle is to adapt downstream tasks to the original training task at minimal cost, thereby utilizing the knowledge embedded in pretrained models to address problems. Prompting was initially introduced in natural language processing (NLP) Liu et al. (2023), where additional tokens are added to token sequences to help the pretrained model better "understand" the task Li & Liang (2021); Lester et al. (2021). Initially, the values of prompt engineering were heuristically selected Brown et al. (2020). Subsequently, prompt methods based on learnable parameters gradually became mainstream due to their efficiency and flexibility Lester et al. (2021); Li & Liang (2021); Liu et al. (2021a); Vu et al. (2021). Due to its simplicity and effectiveness, prompt tuning has also been applied to some visual tasks such as image classification Bahng et al. (2022); Jia et al. (2022), segmentation Nie et al. (2023), and 3D point clouds Wang et al. (2022); Tang et al. (2024); Zhu et al. (2023). To the best of our knowledge, there has been no prior work utilizing prompt tuning in the field of event cameras. We explore the use of prompt tuning for the first time in the task of event camera data pre-training, aiming to achieve an efficient transfer of image pre-training knowledge to event camera data.

## 3 METHOD

### 3.1 OVERVIEW

In Figure 2, we present the overall framework of the SpatioTemporal Information Fusion Prompting (STP). The process consists of three key steps. First, the event stream data is converted into a SpatioTemporal Event Count Image (STECM) that preserves both temporal and spatial information (Section 3.2). Next, the STP progressively fuses the temporal and spatial information from the

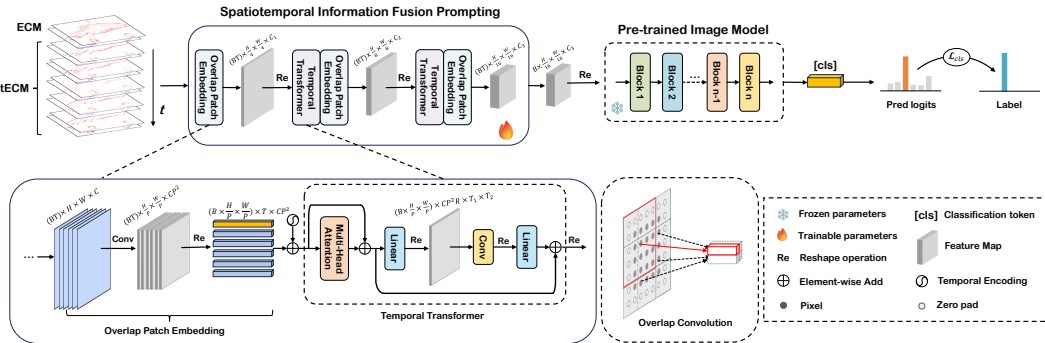

Figure 2: **Framework of STP**. First, the event stream data is converted into tECM and ECM, which are concatenated to form STECM. Then, STECM is fused using STP to generate the spatiotemporal representation $z$. STP consists of two key components: Overlap Patch Embedding and the Temporal Transformer, where Overlap Patch Embedding is primarily implemented using Overlap Convolution. Finally, the spatiotemporal representation $z$ is fed into a frozen pretrained image model for classification, with the classification loss $\mathcal{L}_{cls}$ guiding the training of STP.

STECM while reducing sparse regions, resulting in a spatiotemporal representation of the event stream (Section 3.3). Finally, this representation is fed into a pretrained image model for high-level semantic prediction. During the pretraining phase, the weights of the image model are frozen, guiding the training of the STP module and enabling knowledge transfer from image to event data. In the finetuning phase, both the STP module and the pretrained image model are trained, promoting bidirectional knowledge flow between event data and image data.

## 3.2 EVENT REPRESENTATION

Combining event streams with deep learning typically requires converting event streams into planar representations. There are currently three main representation methods: Event Count Image (ECM) Maqueda et al. (2018); Zhu & Yuan (2018), Voxel grid Zhu et al. (2019), and Event Spike Tensor (EST) Gehrig et al. (2019), with their characteristics and descriptions detailed in Table 1. For event camera pretraining, we aim for a representation that preserves as much of the event stream's characteristics and information as possible, without introducing additional information that could affect its generalizability. However, as observed, none of the current mainstream methods fully retain all the key features of event streams, including temporal information, spatial count information, and event polarity, while avoiding the introduction of extraneous information.

To address this, we propose a novel representation method called the SpatioTemporal Event Count Image (STECM), which incorporates temporal information into the existing ECM representation. This effectively resolves the issue of discarding time information. Specifically, following the approach of the Voxel grid, we divide the event stream into $T$ temporal segments. For each segment of the event stream $\mathcal{E}_T = \{e_k\}_{k=1}^n$, where $n$ is the number of events, each event $e_k$ is represented by a tuple $(x_i, y_i, t_i, p_i)$, where $x_i$ and $y_i$ denote the pixel coordinates, $t_i$ represents the timestamp of the event, and $p_i = \pm 1$ represents the polarity. As event cameras asynchronously report changes in pixel intensity, the output is a series of independent events Brandli et al. (2014); Gallego et al. (2020). To obtain a 2D image structure with visible edges, we accumulate events of different polarities separately onto a 2D plane, resulting in a $ECM \in \mathbb{R}^{2 \times H \times W}$. As illustrated in Figure 2, by concatenating these $T$ ECM, we obtain a temporal ECM representation (tECM) of the event stream. To preserve the spatial characteristics, we also convert the entire event stream $\mathcal{E}$ into a single ECM and concatenate it with the tECM, resulting in the $STECM \in \mathbb{R}^{2 \times (T+1) \times H \times W}$. This representation method retains the complete spatial structure and effective temporal information of the event stream without introducing additional information or constraints. As a result, it can be effectively transferred to a variety of downstream tasks.

| Representation | Dimensions | Description | Characteristics |
|---|---|---|---|
| Event count image (ECM) | $2 \times H \times W$ | Image of event counts | Discards time stamps |
| Voxel grid | $T \times H \times W$ | Voxel grid summing event polarities | Discards event polarity |
| Event Spike Tensor (EST) | $2 \times T \times H \times W$ | Sample event point-set into a grid | Introduced a learnable constraint |
| STECM (our work) | $2 \times (T+1) \times H \times W$ | Segmentation and summarization of event | Preserved spatiotemporal information |

Table 1: **Comparison of Different Event Data Representation Methods.** $T$, $H$, and $W$ represent the temporal dimension of the event stream, image height, and image width, respectively. The descriptions in this table are primarily based on Gehrig et al. (2019).

### 3.3 SPATIOTEMPORAL INFORMATION FUSION PROMPTING

The design of SpatioTemporal Information Fusion Prompting (STP) aims to achieve two key objectives: (i) to compress and integrate the temporal information of event data into spatial information as much as possible, allowing it to align with pretrained image models. Accoraging to the characteristics of event cameras, event streams from different time segments contain non-overlapping yet similar spatial information, providing a foundation for compressing them onto the same plane. (ii) When encoding local information into embeddings, a larger receptive field is needed to reduce sparse areas and avoid overfitting caused by sparsity. Additionally, we must adhere to the principles of prompt tuning by minimizing computational complexity and parameter counts.

Therefore, our proposed STP consists of two components: first, an Overlap Patch Embedding module utilizing overlap convolution with a larger patch window, allowing adjacent patches to share information with the current patch's embedding. This enhances local information exchange and effectively reduces sparse regions. Second, a Temporal Transformer facilitates the interaction and fusion of global and local temporal information. To further minimize computational costs while achieving gradual integration of spatial local information, we employ a pyramid structure design based on pVIT Wang et al. (2021b). This approach progressively enlarges the patch's receptive field and continuously merges information across different temporal dimensions, ensuring thorough fusion of spatiotemporal information.

#### 3.3.1 OVERLAP PATCH EMBEDDING

In the Overlap Patch Embedding, the kernel size of the convolution must be larger than the stride to utilize information from neighboring patches and fill in sparse regions within the current patch. Specifically, given an input event image with dimensions $(BT) \times H \times W \times C$, where $B$ represents batch size, we apply a convolution layer with a stride of $P$ and a kernel size $K$, where $K > P$. The size of the output feature map is $(BT) \times \frac{H}{P} \times \frac{W}{P} \times CP^2$. Next, we apply a $3 \times 3$ convolution to extract finer local details, while keeping the feature map size unchanged. Finally, the feature map is reshaped to size $\left( B \times \frac{H}{P} \times \frac{W}{P} \right) \times T \times CP^2$ and fed into the Temporal Transformer, which fuses temporal information within each patch. The Overlap Patch Embedding is applied three times within the STP, with the kernel size $\{k_1, k_2, k_3\}$ being a set of hyperparameters. The strides are set to $\{4, 2, 2\}$. By progressively increasing the receptive field, each patch is enriched with information from neighboring patches, thereby enhancing its embedding representation and mitigating the risk of overfitting due to sparsity.

#### 3.3.2 TEMPORAL TRANSFORMER

As shown in Figure 2, the Temporal Transformer primarily follows the Transformer block design of ViT Dosovitskiy et al. (2020) but with two key differences. First, instead of introducing an additional token for temporal information aggregation, we use the token corresponding to the ECM as the main token and fuse the tECM's temporal information into it. This is because the ECM contains the primary spatial structure information of the event stream, and with only two Temporal Transformer layers, it is insufficient to aggregate both temporal and spatial information into a new token effectively. The second difference is that we introduce a local temporal aggregation layer between the two linear layers, which enhances local information exchange. Therefore, our Temporal Transformer can be represented by the following equation:

$$\boldsymbol{z}_{l-1} = \left[ \boldsymbol{x}_{ECM};\ \boldsymbol{x}^1_{tECM}; \dots; \boldsymbol{x}^T_{tECM} \right] + \mathbf{E}_{tem}, \quad \boldsymbol{x} \in \mathbb{R}^{1 \times CP^2},\ \mathbf{E}_{tem} \in \mathbb{R}^{(T+1) \times CP^2} \quad (1)$$

$$z_l' = MHA\left(LN\left(z_{l-1}\right)\right) + z_{l-1} \tag{2}$$

$$z_l = Linear\left(Conv\left(Linear\left(LN\left(z_l'\right)\right)\right)\right) + z_l' \tag{3}$$

In Equation 1, $z$ represents the embedding of the STECM, $x$ denotes each temporal token, and $\mathbf{E}_{tem}$ refers to the learnable temporal encoding. In Equation 2, $MHA$ and $LN$ stand for Multi-Head Attention and LayerNorm Dosovitskiy et al. (2020), respectively. In Equation 3, $Linear$ refers to the linear layer. After passing through the first linear layer, the dimension of $z$ becomes $(B \times \frac{H}{P} \times \frac{W}{P}) \times T \times CP^2R$, which is then reshaped into $(B \times \frac{H}{P} \times \frac{W}{P}) \times CP^2R \times T_1 \times T_2$, where $R$ denotes MLP ratio and $T = T_1 \times T_2$. We then apply a convolutional layer ($Conv$) for local temporal aggregation while keeping the shape unchanged. Finally, $z$ is reshaped back and another linear layer is applied to adjust the dimension to $(B \times \frac{H}{P} \times \frac{W}{P}) \times T \times CP^2$.

### 3.4 Training Objectives

**Pretraining**. In the pretraining phase, we utilize ViT Dosovitskiy et al. (2020) as the backbone of our method and **freeze its weights**. We remove the position embedding layer from ViT because the embedding has already been completed in the STP process. The embedding features $\hat{z} \in \mathbb{R}^{B \times (\frac{H}{16} \times \frac{W}{16}) \times C_t}$ from the event stream are input into the pretrained model, resulting in a class token feature $f_{cls} \in \mathbb{R}^{1 \times C_t}$, where $C_t$ represents the token feature dimension. Finally, the class token is used for target classification, optimized by a CrossEntropy loss $\mathcal{L}_{cls}$. This approach effectively transfers the knowledge from the frozen image pretrained model into the STP, guiding it in learning event data features.

**Finetuning**. During the finetuning stage, both the weights of the STP and ViT need to be trained to better adapt to downstream tasks. The classification head can be adjusted based on the specific types of data in the downstream tasks, while the other structures remain unchanged. This approach allows for effective bidirectional knowledge transfer between event data and image, enabling the model to adapt efficiently to various downstream tasks.

## 4 Experiments

### 4.1 Dataset and Experimental Setup

**Pre-training Dataset**. We utilize the N-ImageNet Kim et al. (2021) and ImageNet-1K Deng et al. (2009) datasets for pre-training. N-ImageNet is currently the largest dataset for event camera classification. It is constructed based on the ImageNet-1K dataset, capturing RGB images displayed on a monitor using a moving event camera. This dataset comprises 1,781,167 samples covering 1,000 object categories. Each event data sample has a resolution of $480 \times 640$. To align with the pretraining model, we resize both event data and RGB images to a resolution of $224 \times 224$.

**Implementation**. We adopt VIT-S/16 as the pretrained classification model, freezing its weights during the pretraining phase. After pretraining, the STP and ViT classification models together form the event camera data pretraining model, which is then fine-tuned to complete downstream tasks. For further implementation details and training hyperparameters, please refer to the **Appendix A.1**.

### 4.2 Object Recognition

In this work, we primarily compare our approach with previous methods pretrained on N-ImageNet: ECDP Yang et al. (2023) and MEM Klenk et al. (2024). Additionally, Huang et al. Huang et al. (2024) proposed a data-efficient method, DMM. However, due to its requirement for longer event stream durations, DMM could only be pretrained on the N-Caltech101 dataset. Furthermore, following the approach of ECDP Yang et al. (2023), we include the results of training from scratch, and transfer learning from pretrained models, for comparison. We conduct performance comparisons and analyses on both the large-scale dataset N-ImageNet and small-scale datasets N-Caltech101 Orchard et al. (2015), N-Cars Sironi et al. (2018), and CIFAR-10-DVS Cheng et al. (2020). For N-Caltech101 and CIFAR-10-DVS, training and testing set partitions were not provided, so we randomly split them to generate training and testing datasets (please refer to the **Appendix A.2**).

| Method | Backbone | Pr. Params | Pr. Epoch | N-ImageNet | | N-Caltech101 | N-Cars | CIF10 |
| --- | --- | --- | --- | --- | --- | --- | --- | --- |
| | | | | acc@1 | acc@5 | | | |
| Training from scratch | | | | | | | | |
| EST Gehrig et al. (2019) | - | 21M | - | 48.93 | - | 68.12 | 90.80 | 62.57 |
| ViT | ViT-S/16 | 22.1M | - | 46.70 | 69.89 | 55.63 | 89.14 | 52.45 |
| ViT | ViT-B/16 | 86.6M | - | 51.23 | 74.50 | 67.11 | 93.09 | 55.15 |
| ResNet | ResNet50 | 25.6M | - | 50.07 | 74.83 | 62.69 | 91.20 | 56.65 |
| Transfer learning from models pretrained on ImageNet Deng et al. (2009) | | | | | | | | |
| N-ImageNet | - | - | - | - | - | 80.88 | 91.48 | 70.36 |
| ViT | ViT-S/16 | 22.1M | 300 | 60.48 | 83.02 | 85.02 | 96.76 | 76.10 |
| ViT | ViT-B/16 | 86.6M | 300 | 62.98 | 84.75 | 86.45 | 97.56 | 77.45 |
| ResNet | ResNet50 | 25.6M | 90 | 57.37 | 80.93 | 86.51 | 97.61 | 73.40 |
| Pretraining on N-ImageNet Kim et al. (2021) + Finetuning | | | | | | | | |
| EST Gehrig et al. (2019) | - | 21M | - | - | - | 86.81 | 94.73 | 73.72 |
| ECDP Yang et al. (2023) | ViT-S/16 | 22.1M | 300 | 64.83 | 86.30 | 87.66 | 97.93 | 78.00 |
| MEM Klenk et al. (2024) | dVAE+VIT | 23.1M | 125 | 57.89 | - | 90.10 | 93.27 | - |
| DMM Huang et al. (2024) | - | 13.5M | 700 | - | - | 88.00 | 97.10 | 78.60 |
| Ours | STP | **2.3M** | 100 | **68.83** | **89.53** | **94.48** | **98.86** | **88.67** |

Table 2: Comparison of object recognition accuracy, trainable parameters during the pretraining stage (Pr. Params), and pre-training epochs (Pr. Epoch) on N-ImageNet, N-Caltech101, N-Cars, and CIFAR-10-DVS datasets. Top-1 (acc@1) and top-5 (acc@5) accuracy are shown on N-ImageNet, while only top-1 accuracy is reported on small-scale datasets. '-' indicates either the result is not reported or not supported by the method. **Bold** and underline indicate the best and second-best results.

**Results on N-ImageNet**. As shown in Table 2, our method achieved a top-1 accuracy of 68.83%, surpassing the previous best result (64.83%) by **4%**. This indicates that our approach effectively leverages pretraining knowledge from the image domain, opening up new avenues for pretrained models in event camera data. It's worth noting that the results of training from scratch were inferior, possibly due to overfitting caused by the sparse nature of event stream data. While transfer learning can partially transfer pretrained knowledge from images, it still fails to fully address the issue of data sparsity. In contrast to other methods that require pretraining on large-parameter backbones such as VIT and ResNet, our STP has only **2.3M trainable parameters** and requires training for only 100 epochs, effectively reducing the demand for training resources.

**Results on small-scale datasets**. As shown in Table 2, our method achieved top-1 accuracies of 94.48%, 98.86%, and 88.67% on the N-Caltech101, N-Cars, and CIFAR-10-DVS datasets, respectively. Compared with the previous SOTA methods, our approach improved by **4.38%**, **0.93%**, and **10.07%**, respectively. This demonstrates the efficient transfer of knowledge from pretrained image models to the event camera domain, highlighting the effectiveness of our method.

## 4.3 ABLATION STUDIES

To validate the effectiveness of our proposed framework and STP, we conducted extensive ablation studies on the N-ImageNet classification task. These studies include evaluations of various event data representation methods, different prompting models, variants of the STP, different pretraining model backbones, as well as model hyperparameter settings.

**Event Data Representation Methods**. We compared different event stream representation methods, including ECM Maqueda et al. (2018), Voxel grid Zhu et al. (2019), and EST Gehrig et al. (2019), against our proposed STECM. To maintain consistency with STECM's dimensions, we appended an additional temporal token to the Voxel grid and EST. For ECM, which lacks a temporal dimension, we replicated it $T$ times and concatenated them to match the dimensions of STECM. As shown in Table 3(a), the Voxel grid exhibited a significant performance drop due to the omission of event polarity. EST, on the other hand, imposed additional constraints on the event stream, reducing its generalization capability. ECM failed to retain temporal information, making it an incomplete representation of the

**(a) Event Representation**

| Representation | Pr. | Ft. |
|---|---|---|
| ECM | 62.86 | 67.78 |
| Voxid grid | 48.69 | 55.26 |
| EST | 63.76 | 67.94 |
| STECM | **64.46** | **68.83** |

**(b) Prompting Model**

| Prompting | #Params | Pr. | Ft. |
|---|---|---|---|
| E2VID | 4.5 M | 52.24 | 58.93 |
| STP-vanilla | 4.1 M | 60.34 | 66.80 |
| STP | **2.3 M** | **64.46** | **68.83** |

**(c) Kernel size of OPE**

| $\{k_1, k_2, k_3\}$ | #Params | Ft. |
|---|---|---|
| $\{6, 4, 4\}$ | **1.2 M** | 68.07 |
| $\{8, 6, 6\}$ | 2.3 M | 68.83 |
| $\{10, 8, 8\}$ | 3.8 M | **68.96** |

**(d) Variants of the STP**

| Model | OPE Overlap | OPE Conv | LTA | Pr. | Ft. |
|---|---|---|---|---|---|
| #A | ✗ | ✓ | ✓ | 63.26 | 66.94 |
| #B | ✓ | ✗ | ✓ | 64.17 | 67.23 |
| #C | ✗ | ✗ | ✓ | 57.70 | 62.46 |
| #D | ✓ | ✓ | ✗ | 63.84 | 68.02 |
| STP | ✓ | ✓ | ✓ | **64.46** | **68.83** |

**(e) Architecture of the pretraining models**

| Pr. Arch | #Params | Pr. | Ft. |
|---|---|---|---|
| ResNet50 | 25.6M | 59.70 | 65.93 |
| ConvNeXt_T | 28.6M | 67.09 | 71.90 |
| Swin_T | 28.3M | 62.58 | 67.34 |
| ViT-S/16 | **22.1M** | 64.46 | 68.83 |
| ViT-B/16 | 86.6M | 73.14 | 75.88 |
| ViT-L/16 | 304.3M | **75.51** | **78.06** |

Table 3: Ablation experiments on N-ImageNet Kim et al. (2021) classification. We report the top-1 accuracy of our method under different ablation conditions in terms of pre-training (Pr.) and fine-tuning (Ft.) stages. (a) Impact of different event data representation methods. (b) Comparison of the performance of different prompting models. (c) Different kernel sizes in Overlap Patch Embedding. (d) Performance of various STP variants. (e) Effect of different image pretraining models. The gray area represents the baseline of STP. Best in **bold**.

event stream. The results demonstrate that our proposed STECM effectively captures the essential features of the event stream, leading to a significant improvement in pretraining performance.

**Prompting Models**. As we are the first to propose a prompting-based approach for event camera data, there are no directly comparable models. To provide a fair comparison, we replaced the STP module with other methods. One approach followed previous practices by reconstructing the event stream into images Sun et al. (2022) and adapting them to a pretrained image model. We used the E2VID Rebecq et al. (2019) method as the reconstruction model. Additionally, we designed a vanilla for event data prompting (STP-vanilla), where the temporal dimension of the event stream was first aggregated using a linear layer and then passed through two transformer blocks, which were identical to the ones used in VIT Dosovitskiy et al. (2020). As shown in Table 3(b), our STP method effectively integrates the spatiotemporal information of the event stream in a progressive manner, resulting in a significant performance improvement compared to other methods.

**Variants of the STP**. Our proposed STP method consists of two key components: Overlap Patch Embedding (OPE) and the Temporal Transformer. We conducted ablation studies to examine different variations of these components: 1) Model#A: We removed the Overlap window mechanism and used independent patch division similar to ViT with position embeddings. 2) Model#B: We eliminated the fine-grained information extraction layer ($Conv$) following the Patch Embedding. 3) Model#C: Both the Overlap window and the fine-grained information extraction layer were removed. 4) Model#D: We removed the local temporal aggregation (LTA) layer from the Temporal Transformer, keeping all other structures intact.

As shown in Table 3(d), maintaining global information exchange while enhancing local information fusion during the prompting process is crucial, significantly improving the integration of event data. Additionally, we examined the impact of the overlap mechanism on model performance, particularly in handling sparse event data. We selected two sparse event images and input them into both overlapping and non-overlapping versions of the STP. After feeding the fused event features into the pretrained image model, we visualized the attention weights from the 1*st*, 6*th*, and 12*th* layers. As depicted in Figure 3, the attention weight matrix of the overlapping STP is much more evenly distributed, while the non-overlapping STP matrix shows more concentrated attention. This indicates that the overlap mechanism effectively fills the sparse regions of event data, leading to a more comprehensive event data representation and thereby mitigating the overfitting issue.

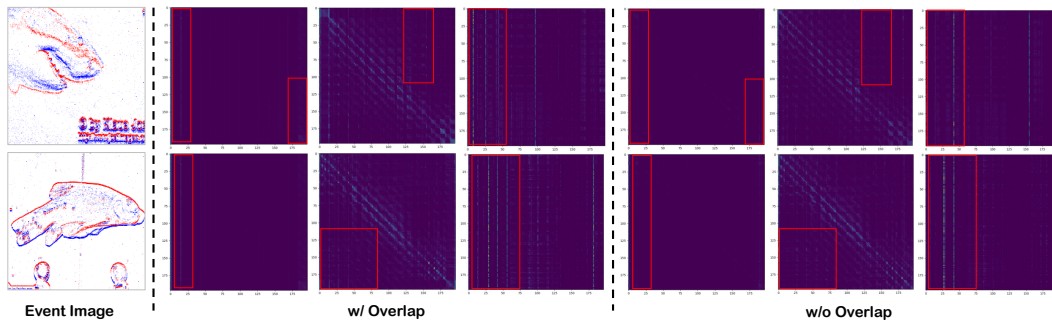

Figure 3: **Visualization of event images and their corresponding attention matrices.** For models w/ Overlap and w/o Overlap, from left to right are attention weights from the 1*st*, 6*th*, and 12*th* layers. The comparison shows that the attention matrix of model w/ Overlap is more uniformly distributed (highlighted in the red box).

**Ablation Study on Pretraining Models**. Our method initially uses VIT-S/16 pretrained on ImageNet. For comparison, we replaced it with three image classification pretrained models with comparable parameter counts: ResNet50 He et al. (2016), Swin-T Liu et al. (2021b), and ConvNeXt-T Liu et al. (2022). Additionally, we experimented with scaling the parameters of VIT, using VIT-B/16 and VIT-L/16. As shown in Table 3(e), our method achieves superior results across various image pre-training models, surpassing the previous SOTA methods. Notably, our approach achieved a top-1 accuracy of 78.06% with VIT-L/16, **marking the first instance of surpassing 75% accuracy on the N-ImageNet**. This result further demonstrates the potential of our method to fully leverage the knowledge embedded in image pretrained models.

**Ablation Study on Model Hyperparameters**. We further explored the impact of the kernel size in the Overlap Patch Embedding on model performance, as this parameter determines the size of the local receptive field during event data encoding. In our previous training, we set $\{k_1 = 8, \ k_2 = 6, \ k_3 = 6\}$. As shown in Table 3(c), the kernel size has a significant effect on the parameter count of the STP model and also influences its performance on downstream tasks. This demonstrates that increasing the local receptive field can effectively alleviate the overfitting caused by data sparsity. For more detailed hyperparameter experiments, please refer to the **Appendix B**.

## 4.4 SEMANTIC SEGMENTATION

We finetuned the pretrained STP together with the image pretraining model on the downstream semantic segmentation task. Following the approach in Bao et al. (2021), we attached the UperNet decoder Xiao et al. (2018); Bao et al. (2021) to our pretrained model to estimate semantic labels. We conducted experiments on the DDD17 Binas et al. (2017); Alonso & Murillo (2019) and DSEC datasets Gehrig et al. (2021); Sun et al. (2022), using mean Intersection over Union (mIoU) as the evaluation metric. To compare with previous methods, we used ResNet50 He et al. (2016) as the image pretraining model and included prior SOTA models such as EV-SegNet Alonso & Murillo (2019) and ESS Sun et al. (2022) in the comparison. Additionally, we integrated Hierarchical Features from STP into the semantic segmentation pipeline via a linear layer (w/ STF). More details on this approach can be found in the **Appendix A.3**. As shown in Table 4(a), our method achieves the best results on both datasets, surpassing the SOTA ESS. Incorporating Hierarchical Features from STP improved the model's ability to capture temporal information in the event stream, further enhancing segmentation performance. Table 4(b) also presents our semantic segmentation results, demonstrating the excellent generalization of our pre-training method. For more details and training parameters, please refer to the **Appendix A.3**.

## 4.5 OPTICAL FLOW ESTIMATION

Event cameras excel at capturing dynamic data, making motion information measurement a crucial downstream task. Following previous approaches, we evaluate our optical flow estimation performance on the MVSEC dataset Zhu et al. (2018). We replaced the classification head in our pretrained

(a) Quantitative Analysis

| Method | Backbone | DDD17 | DSEC |
|---|---|---|---|
| Training from scratch | | | |
| EV-segnet | - | 54.81 | 51.76 |
| ResNet | ResNet50 | 56.96 | 57.60 |
| Transfer learning | | | |
| ESS | - | 61.37 | 53.29 |
| ResNet | ResNet50 | 59.25 | 58.50 |
| Pre-training on N-ImageNet | | | |
| ECDP | ResNet50 | 59.15 | 59.16 |
| MEM | dVAE+VIT | - | 44.62 |
| DMM | - | 60.59 | 58.78 |
| Ours | STP | **61.98** | **61.07** |
| Ours (w/ STP) | STP | **62.13** | **61.29** |

(b) Examples of semantic segmentation on the DSEC dataset. Columns 1/4 show event images (blue for positive events, red for negative events), columns 2/5 show segmentation results, and columns 3/6 show the ground truth.

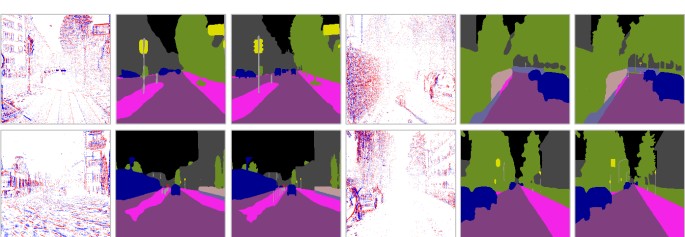

Table 4: Semantic segmentation comparison on DDD17 and DSEC datasets. We report the mean intersection over Union (mIoU, %) for each dataset. (a) Quantitative comparison of semantic segmentation results. (b) Visualization of semantic segmentation results.

| Method | Backbone | indoor_flying1 | | indoor_flying2 | | indoor_flying2 | |
|---|---|---|---|---|---|---|---|
| | | AEE | Outlier | AEE | Outlier | AEE | Outlier |
| Previous SOTA method | | | | | | | |
| EST Gehrig et al. (2019) | - | 1.24 | 5.09 | 2.05 | 19.90 | 1.71 | 11.67 |
| DCEFlow Wan et al. (2022) | - | 0.75 | 0.60 | 1.39 | 8.01 | 1.13 | 5.29 |
| Transfer learning from models pretrained on ImageNet Deng et al. (2009) | | | | | | | |
| ViT | ViT-S/16 | 0.88 | 3.06 | 1.79 | 16.63 | 1.49 | 8.66 |
| ResNet | ResNet50 | 0.60 | 0.23 | 1.37 | 8.76 | 1.15 | 5.34 |
| Pretraining on N-ImageNet Kim et al. (2021) + Finetuning | | | | | | | |
| ECDP Yang et al. (2023) | ResNet50 | 0.6 | 0.35 | 1.35 | 8.57 | 1.12 | 5.26 |
| ECDP Yang et al. (2023) | ViT-S/16 | 0.61 | **0.05** | 1.26 | 6.69 | 1.00 | 3.11 |
| Ours | ViT-S/16 | **0.58** | **0.05** | **1.22** | **6.34** | **0.93** | **3.03** |

Table 5: Comparison of optical flow estimation on the MVSEC dataset Zhu et al. (2018) using different methods. The evaluation is based on Average Endpoint Error (AEE) and Outlier Percentage (%), following the KITTI benchmark Menze et al. (2015).

network with a decoder network to estimate optical flow He et al. (2022); Bao et al. (2021). More details can be found in the **Appendix A.4**. As shown in Table 5, our method, utilizing ViT-S/16 as the backbone, achieves the lowest Average Endpoint Errors (AEE) and outlier ratios (Outlier) compared to other methods. Additionally, we provide example visualizations of optical flow predictions in the Figure 6. These results further demonstrate the strong generalization capability of our approach across different tasks.

## 5 CONCLUSION

In this paper, we propose a Spatiotemporal Information Fusion Prompting (STP) method that effectively bridges the gap between event stream data and pretrained image models, facilitating efficient knowledge transfer. Our approach progressively fuses the spatiotemporal information in event data to generate representations compatible with image pretrained models, enabling seamless transfer of knowledge from images to events. Additionally, by expanding the receptive field and enhancing local information fusion, we address the sparsity issue inherent in event data, achieving bidirectional knowledge transfer between event streams and images. Experimental results demonstrate the superiority and potential of our pretraining method, offering a novel perspective for pretraining models on event camera data.

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

APPENDIX

# A EXPERIMENT SETTINGS

## A.1 PRE-TRAINING

Our pretraining setup primarily follows the methodology outlined in previous work Yang et al. (2023). The hyperparameters are detailed in Table 6(a). Specifically, the learning rate is linearly scaled with the batch size, i. e., lr = base lr × batch size / 256.

Table 6: Hyperparameters for pretraining (a) and for finetuning on the object recognition task (b).

### (a) Pre-training

| Hyperparameters | Value |
|---|---|
| optimizer | AdamW |
| base lr | $1.5 \times 10^{-4}$ |
| weight decay | $3 \times 10^{-2}$ |
| batch size | 512 |
| epochs | 100 |
| warmup epochs | 20 |
| lr scheduler | cosine |
| label smoothing | 0.8 |

### (b) Fine-tuning on the object recognition task

| Hyperparameters | N-ImageNet | N-Caltech101 | N-Cars | CIF10 |
|---|---|---|---|---|
| optimizer | AdamW | AdamW | AdamW | AdamW |
| base lr | $1 \times 10^{-4}$ | $2.5 \times 10^{-4}$ | $1.25 \times 10^{-4}$ | $2.5 \times 10^{-4}$ |
| weight decay | $1 \times 10^{-1}$ | $5 \times 10^{-2}$ | $5 \times 10^{-2}$ | $3 \times 10^{-1}$ |
| batch size | 256 | 512 | 512 | 512 |
| epochs | 50 | 100 | 100 | 100 |
| warmup epochs | 10 | 20 | 20 | 20 |
| lr scheduler | cosine | cosine | cosine | cosine |
| gradient clipping | 5 | 5 | 5 | 5 |
| drop path rate | $1 \times 10^{-1}$ | $1 \times 10^{-1}$ | $1 \times 10^{-1}$ | $1 \times 10^{-1}$ |

## A.2 OBJECT RECOGNITION

We fine-tuned our STP on the N-ImageNet Kim et al. (2021), N-Caltech101 Orchard et al. (2015), N-Cars Sironi et al. (2018), and CIFAR-10-DVS Cheng et al. (2020) datasets to evaluate its performance on the object recognition task (Table 6(b)). For the N-Caltech101, N-Cars, and CIFAR-10-DVS datasets, we adjusted the final classification head of the VIT model to match the number of classes in these datasets. Additionally, since the N-Caltech101 and CIFAR-10-DVS datasets do not have predefined training and testing splits, we followed previous work Yang et al. (2023) and randomly split these datasets, using 80% for training and 20% for testing.

## A.3 SEMANTIC SEGMENTATION

[h] For the semantic segmentation task, we embedded the UperNet decoder Xiao et al. (2018); Bao et al. (2021) into the pretrained model and fine-tuned it alongside STP on the dataset. We trained using cross-entropy and Dice loss Sudre et al. (2017), and evaluated performance with the mean Intersection over Union (mIoU) metric. Table 7 shows our finetuning hyperparameters. We present more semantic segmentation results on the DSEC dataset in Figure 4.

Table 7: Fine-tuning hyperparameters on the DDD17 Binas et al. (2017) and DSEC Gehrig et al. (2021) datasets.

| Hyperparameters | DDD17 | DSEC |
|---|---|---|
| optimizer | AdamW | AdamW |
| lr | $1 \times 10^{-3}$ | $1 \times 10^{-3}$ |
| weight decay | $5 \times 10^{-2}$ | $5 \times 10^{-2}$ |
| batch size | 32 | 32 |
| epochs | 100 | 100 |
| warmup epochs | 10 | 10 |
| lr scheduler | cosine | cosine |
| gradient clipping | 3 | 3 |
| drop path rate | $1 \times 10^{-1}$ | $1 \times 10^{-1}$ |

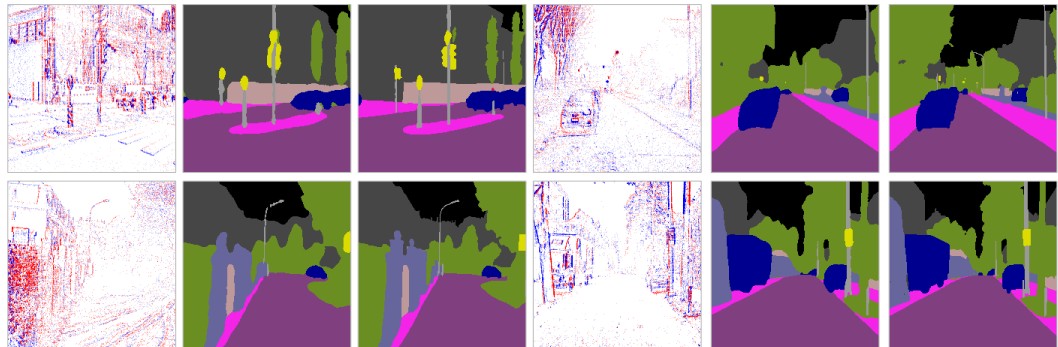

Figure 4: Examples of semantic segmentation on the DSEC dataset. Columns 1/4 show event images, columns 2/5 show segmentation results, and columns 3/6 show the ground truth.

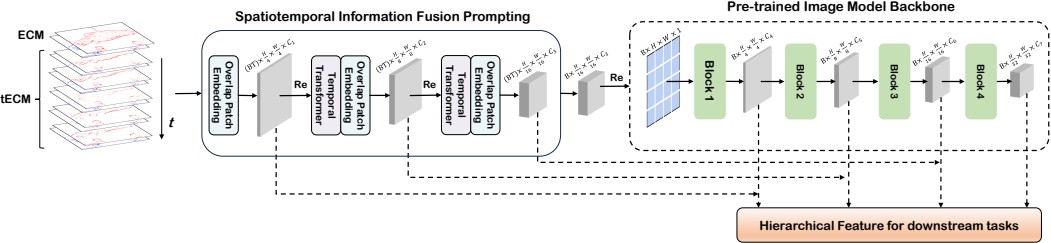

Figure 5: The framework for utilizing the Hierarchical Features from STP for semantic segmentation.

Additionally, in STP, the model generates hierarchical features, which can be utilized for semantic segmentation tasks. To leverage these features, we apply a linear projection layer to transform them into the same embedding dimension and connect them to the backbone (w/ STP). The specific implementation is illustrated in Figure 5. As shown in Table 4(a), this approach effectively provides more detailed temporal information, significantly improving the performance of semantic segmentation.

## A.4 OPTICAL FLOW ESTIMATION

We attached a UperNet decoder Xiao et al. (2018); Bao et al. (2021) to our pretrained network for optical flow estimation. Additionally, inspired by previous work Yang et al. (2023), we added a patch embedding layer as used in Yue et al. (2021) to the ViT. To accomplish this, we first reshape the spatiotemporal representation $z$ into $B \times C \times H \times W$, which is then fed into the embedding layer. We use the $L1$ loss for supervision and train using the MVSEC dataset Zhu et al. (2018) setup defined by Yang et al. (2023). Detailed optimization settings can be found in Table 8. The visual results of the optical flow estimation can be seen in Figure 6.

Table 8: Fine-tuning hyperparameters on the MVSEC Zhu et al. (2018) datasets.

| Hyperparameters | MVSEC |
| --- | --- |
| optimizer | AdamW |
| lr | $1 \times 10^{-3}$ |
| weight decay | $1 \times 10^{-4}$ |
| batch size | 256 |
| epochs | 150 |
| warmup epochs | 20 |
| lr scheduler | cosine |
| gradient clipping | none |
| drop path rate | $1 \times 10^{-1}$ |

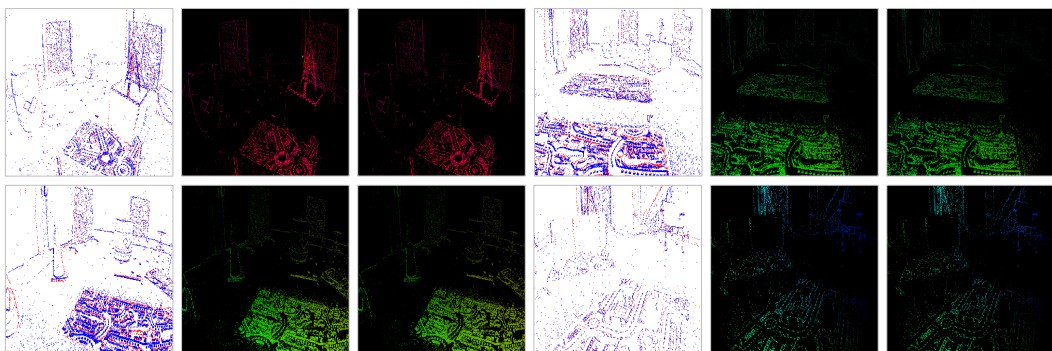

Figure 6: Visualization of the optical flow estimation results on MVSEC dataset. Columns 1/4 show event images, columns 2/5 show optical flow estimation results, and columns 3/6 show the ground truth.

## B  ABLATION STUDIES

**Ablation Studies on Token Selection for Spatiotemporal Representation**. For the spatiotemporal representation $z$ input to the pretrained image model, we experimented with different token selection strategies: adding an extra token (Add), using the token corresponding to the ECM (ECM), or averaging all tokens (Avg). As shown in Table 9(a), using the token corresponding to the ECM as the spatiotemporal representation $z$ effectively captures the event stream's spatiotemporal information, leading to improved model performance.

**Ablation Studies on Number of Event Stream Segments** $T$. Segmenting the event stream effectively preserves its temporal information. However, increasing the number of segments also increases the computational cost, impacting the model's runtime performance. Following the approach used in Voxid grid Zhu et al. (2019), we set $T = 5$. Additionally, we explored the impact of different values of $T$ on STP performance. The results are shown in Table 9(b).

Table 9: Ablation studies on the Token Selection and Number of Event Stream Segments $T$.

(a) Ablation of Token Selection

| Token Selection | Pr. | Ft. |
|---|---|---|
| Add | 63.97 | 68.58 |
| Avg | 64.11 | 68.61 |
| ECM | **64.46** | **68.83** |

(b) Ablation of $T$

| $T$ | Pr. | Ft. |
|---|---|---|
| 3 | 64.14 | 68.61 |
| 5 | 64.46 | 68.83 |
| 7 | **64.49** | **68.92** |

