# OpenReview forum: "Enhancing Event Camera Data Pretraining via Prompt-Tuning with Visual Models"
_ICLR.cc/2025/Conference — ICLR 2025 Conference Withdrawn Submission_

### Official Review · Reviewer_Vroc · 2024-10-19

**Soundness:** 2
**Presentation:** 2
**Contribution:** 2
**Rating:** 3
**Confidence:** 4

**Summary:**

This paper proposes STP, a prompt-tuning method for model pre-training for event camera data. STP converts raw event streams into regular spatio-temporal grids, fuses their information, and input to a pre-trained model from the RGB domain. The core idea of this work is that both spatial and temporal information are crucial for event-based perception tasks. Thus, the authors design a Overlap Patch Embedding module for learning spatial interactions, and a Temporal Transformer for learning temporal interactions. Experiments show strong results on classification, semantic segmentation, and optical flow estimation tasks.

**Strengths:**

- Leveraging pre-trained models from RGB domain for event camera data is an important direction, as event datasets are all relatively small. Prompt tuning is a promising direction for label-efficient model adaptation.
- The spatio-temporal fusion design is intuitive and reasonable. Ablations in Table 3. validate the design choice.
- The performance on N-ImageNet is impressive. Results on semantic segmentation and optical flow estimation also beat prior works, though I have concern about the baseline selection (see Weakness)

**Weaknesses:**

1. My main concern is that this paper seems to ignore a very important baseline: ECDDP [1]. This work is published at ECCV, and appeared on arXiv in 2023. It also works on the event camera data pre-training task with similar downstream task evaluations. I think it should definitely be discussed and compared to.
- In fact, the semantic segmentation performance of this work is not higher than ECDDP. In addition, the optical flow estimation performance of this work is much worse.

[1] Yang, Yan, Liyuan Pan, and Liu Liu. "Event Camera Data Dense Pre-training." ECCV. 2024.

2. I also have a question about the designed event representation. The authors claim that their SpatioTemporal Event Count Image (STECM) is novel, but I think similar design has already appeared in prior works. For example, the spatio-temporal voxel grid representation in [2], where they also merge events within small temporal bins, and process positive and negative polarity separately.
3. In addition, I am curious about why you need to convert the entire event stream into one ECM and concatenate it with tECM. Isn't this information redundant? Because this global ECM is just the sum of all tECM.

[2] Zou, Yunhao, et al. "Learning to reconstruct high speed and high dynamic range videos from events." CVPR. 2021.

4. The Overlap Patch Embedding + Temporal Transformer design is interesting, but I wonder how much improvement it brings compared to more standard operations, e.g., factorized spatio-temporal attention (first do 2D spatial attention, then do 1D temporal attention). To save memory, you can use Swin-Transformer style attention for the spatial attention. An ablation would be appreciated.

5. Line 303 says "We utilize the N-ImageNet and ImageNet-1K datasets for pre-training". Where are RGB images used in the proposed framework? From Fig.2 I think STP only takes in event, no RGB image?

6. Line 309 says "We adopt VIT-S/16 as the pretrained classification model". No detail is given about this pre-trained ViT model. Is it supervised pre-trained in the classification task? Is it taken from Huggingface / torchhub or anywhere else? If it is supervised pre-trained, I feel it becomes a unfair comparison with baselines. I believe the backbone from MEM and ECDDP is randomly initialized, and ECDP is self-supervised pre-trained, not supervised.

7. Line 406 claims STP is "the first to propose a prompting-based approach for event camera". I doubt if this is true. For example, EventCLIP [3] prompts a pre-trained CLIP model to perform classification task as well. Of course, it is under a different setting (it uses text input). I would still encourage the author to discuss this work as it is relevant.

[3] Wu, Ziyi, Xudong Liu, and Igor Gilitschenski. "Eventclip: Adapting clip for event-based object recognition." arXiv preprint arXiv:2306.06354 (2023).

**Questions:**

See Weakness for main questions

Minor: the paper seems to use a wrong citation command in Latex. Please use \citep{} so that there will be a bracket around the citations. Otherwise it is very hard to distinguish between citations and main texts.

---

### Official Review · Reviewer_gHzb · 2024-10-29

**Soundness:** 2
**Presentation:** 2
**Contribution:** 2
**Rating:** 5
**Confidence:** 4

**Summary:**

The paper proposes adding a temporal transformer with overlapping patch embedding on top of a frozen and pre-trained RGB image model. The model is pre-trained using both event data and images before fine-tuning on downstream tasks. An event representation called  spatiotemporal event count image is also proposed, based on the the voxel grid.

**Strengths:**

The method achieves state-of-the-art performance across multiple downstream tasks, with particularly high performance on the N-ImageNet dataset (Table 3e).

**Weaknesses:**

Main Concern: The paper primarily adds additional components to a pre-trained image model, using both event and RGB images to train these components, which function as an adapter. Numerous related works in the parameter-efficient fine-tuning domain could serve as comparisons here.  For example, [1]. Additionally, compared to using the ViT model alone for downstream tasks such as classification, what is the additional computational cost such as FLOPs introduced by the adapter?

Other Comments:

* A main critique of previous methods is ignoring the temporal information in events. However, in Table 3a, the performance of ECM and STECM are similar. If temporal information is crucial for object classification in event data, why does the Voxel grid perform lower than ECM? More analysis on the role of polarity and temporal information would be beneficial, as the contribution of STECM to ECM appears incrementally minor. Visualizing the output embeddings of the spatiotemporal information fusion could be helpful for the analysis.

* Why is an adapter specifically introduced to "predict" the patch embedding for the pre-trained image model? Could alternative approaches, such as embedding prediction at different model blocks, be explored?

* During pre-training, is the label based on ground truth, or is it a pseudo-label, such as predicted logits from the image? The information is unclear.

* The proposed method may not be limited to pre-trained classification models and could potentially be applied to other pre-trained models, such as those for segmentation. However, I could not find related experiments to support the applicability.

[1] AdaptFormer: Adapting Vision Transformers for Scalable Visual Recognition

**Questions:**

Please see the weaknesses.

---

### Official Review · Reviewer_yCX8 · 2024-10-31

**Soundness:** 2
**Presentation:** 3
**Contribution:** 2
**Rating:** 6
**Confidence:** 5

**Summary:**

This paper focus on the event camera data pretraining task, and conclude the challenges of applying existing pretraining-finetuning paradigm to the sparse event data domain. Also, this paper strengthen the spatil-temporal characteristics of event data. To address the summarized problems, this paper introduces a low-parameter-cost SpatioTemporal Information Fusion Prompting method which enables bidirectional fusion of event and image data while mitigating the risk of overfitting. Moreover, an Overalp Patch Embedding module is proposed to employ wide receptive field to capture more local information and reduce the influence of sparse regions. Additionally, a Temporal Transformer is introduced to integrate both global and local information. The proposed method significantly outperforms previous approaches.

**Strengths:**

1. Clear summary of existing key challenges for event camera data pretraining.
2. Clear writing, logic flow, and good content arrangement.

**Weaknesses:**

The weaknesses of the paper can be summarized as follows:

Omission of Key Related Works: Several important related works are not adequately discussed. These include EventBind, EventDance, and ExACT, all of which address significant issues in the event camera domain, such as the dataset scarcity and cross-modal knowledge transfer from the RGB domain. The paper lacks a thorough comparison of how the proposed approach aligns with or diverges from these state-of-the-art methods, particularly in areas like representation and conceptual reasoning.

Unclear Connection Between Challenges and Solutions: While the introduction effectively outlines key challenges in event camera data pretraining, it fails to establish a clear link between these challenges and the proposed solutions. This disconnect raises questions about the efficacy of the proposed modules in addressing the identified challenges.

Limited Comparison with Recent State-of-the-Art (SoTA) Methods: The experiments section does not include comparisons with the latest SoTA methods, such as EventBind, limiting the evaluation of the proposed method’s competitiveness. Including these recent approaches would provide a more rigorous benchmark for assessing performance.

**Questions:**

1. [Introduction] "These events are sorted as positive or negative depending on whether the light intensity increases or decreases." Is there any threshold to sort the positive and negative events?
2. [Introduction] Important related works are not discussed, including
[1] EventBind: Learning a Unified Representation to Bind Them All for Event-based Open-world Understanding, ECCV 2024,
[2] EventDance: Unsupervised Source-free Cross-modal Adaptation for Event-based Object Recognition, CVPR 2024,
All these works aim at solving the lack of event camera dataset problem and transferring knowledge from the RGB domain. Please give clear discussion regarding these works.
3. [Introduction] Important related works are not discussed, including
[3] ExACT: Language-guided Conceptual Reasoning and Uncertainty Estimation for Event-based Action Recognition and More, CVPR 2024.
The ExACT work also introduces a similar event representation to maintain the spatiao-temporal characteristic of event data, please give the technical comparison between ExACT and the proposed STECM.
4. [Introduction] The summarized key challenges for event camera data pretraining is clear, however, how do the authors solve these challenges with their proposed method is not clear. Please show how these proposed modules solve the summarized challenges.
5. [Related Works] Important related works are not discussed, including
[1] EventBind: Learning a Unified Representation to Bind Them All for Event-based Open-world Understanding, ECCV 2024,
[2] EventDance: Unsupervised Source-free Cross-modal Adaptation for Event-based Object Recognition, CVPR 2024,
All these works aim at solving the lack of event camera dataset problem and transferring knowledge from the RGB domain. Please give clear discussion regarding these works.
[3] ExACT: Language-guided Conceptual Reasoning and Uncertainty Estimation for Event-based Action Recognition and More, CVPR 2024.
6. [Method] Why the segmentation and summarization of event can preserve the spatio-temporal information? There is still information loss when segmenting and summarizing events. The best choice may be the learning-based EST, please give discussion regarding this problem.
7. [Experiments] The recent SoTA methods are not in the comparison table, such as EventBind.
8. [Ablation] There are only 3 compared event representations, more recent ones should be included, such as ExACT.

---

### Official Review · Reviewer_o9pu · 2024-11-04

**Soundness:** 2
**Presentation:** 2
**Contribution:** 2
**Rating:** 3
**Confidence:** 3

**Summary:**

This paper introduces a novel pretraining approach for event camera data using a method called SpatioTemporal Information Fusion Prompting (STP).
STP integrates spatial and temporal data from event streams and enables knowledge transfer from image-based models.
The authors demonstrate that this method outperforms prior models on tasks like classification, segmentation, and optical flow estimation.

**Strengths:**

1. The paper is well-written and well-organized.
2. The proposed method shows better results in various tasks, compared to previous methods.

**Weaknesses:**

1. The novelty is limited. The SpatioTemporal Information Fusion Prompting appears to be a combination of existing methods. The prompt tuning is similar to P2P: Tuning Pre-trained Image Models for Point Cloud Analysis with Point-to-Pixel Prompting.
2. The paper lacks a comprehensive ablation study of different pretraining and fine-tuning methods, such as LoRA and unfreezing the pretrained model during pretraining, etc.

**Questions:**

see the Weaknesses section.

---

### Note · Authors · 2024-11-13

I have read and agree with the venue's withdrawal policy on behalf of myself and my co-authors.